# LAtte: Hyperbolic Lorentz Attention for Cross-Subject EEG Classification

## Abstract

Electroencephalogram (EEG) classification is critical for applications ranging from medical diagnostics to brain-computer interfaces, yet it remains challenging due to the inherently low signal-to-noise ratio (SNR) and high inter-subject variability. To address these issues, we propose LAtte, a novel framework that integrates a Lorentz Attention Module with an InceptionTime-based encoder to enable robust and generalizable EEG classification. Unlike prior work, which evaluates primarily on single-subject performance, LAtte focuses on cross-subject generalization. First, we learn a shared baseline signal across all subjects using pretraining tasks to capture common underlying patterns. Then, we utilize low-rank adapters to learn subject-specific embeddings that model individual differences. This allows a single model to be trained concurrently for all subjects without maintaining separate sets of weights for each individual, and enables generalization to unseen subjects. We evaluate LAtte on three well-established EEG datasets, achieving a substantial improvement in performance over current state-of-the-art methods.

## 1 Introduction

Electroencephalogram (EEG) classification is a key task in time-series analysis, both from a theoretical and practical perspective, due to its inherent complexity and significant challenges (Parbat & Chakraborty, 2021; Pan et al., 2022). Most significantly, EEG sequences are often contaminated with various artifacts, including eye blinks, eye movements, and muscle activity (Kotte & Dabbakuti, 2020; Delorme, 2023), which can significantly degrade the quality of the recorded data. These artifacts obscure relevant neural signals and reduce the signal-to-noise ratio (Johnson, 2006), thus affecting the performance of downstream applications such as classification and clinical analysis.

Another challenge that arises from these artifacts is the diversity and complexity of noise sources, each with distinct characteristics, which exacerbates the initial issue (Eltrass & Ghanem, 2021). Ocular artifacts (Croft & Barry, 2000), including eye movements and blinks, produce large voltage fluctuations. Myogenic artifacts (Muthukumaraswamy, 2013; Pion-Tonachini et al., 2019), caused by facial and scalp muscle movements, introduce high-frequency noise that overlaps with the brain signal frequency bands. This compounded noise makes it difficult to isolate the task-relevant signal from it and leads to suboptimal performance in classification models.

Previous approaches to EEG classification have explored geometric methods (Lotte et al., 2007), deep learning techniques (Roy et al., 2019). Another notable line of work is Riemannian Learning, with a prominent example being Matt, which models EEG data as symmetric positive-definite (SPD) matrices on a Riemannian manifold (Pan et al., 2022). More recently, Bdeir et al. (2025) introduced Hypermatt, which extends the Matt formulation by replacing the SPD decoder with a hyperbolic variant while keeping the original encoder.

In this work, we take a step further by developing a fully hyperbolic model that leverages hyperbolic geometry in both the encoder and decoder. Hyperbolic manifolds are naturally suited for representing hierarchical structures, a property extensively focused on in the Graph Neural Network literature (Mettes et al., 2024). EEG data inherently exhibits a spatial hierarchy due to the sensor arrangement and node signal interactions, making this setting particularly appropriate.

By capturing these hierarchies and modeling signal interactions directly in Lorentz space, our approach learns more expressive latent representations and achieves cross-subject generalization without requiring a separate model for each patient.

This is particularly important because the low signal-to-noise ratio (SNR) in EEG data poses a significant challenge when training a single joint model across all subjects. The model must contend with highly variable, subject-specific noise distributions. Consequently, most previous work on small EEG datasets, the most common and practical setting, trains separate models per subject. While this can improve within-subject performance, it has severe limitations: it fails to leverage shared EEG patterns across the larger feature space of the entire dataset, and it requires individual model training and tuning for every subject.

We address this by training our proposed model LAtte jointly across all subjects while incorporating subject-specific low-rank adapters (LoRA) (Hu et al., 2022) for subject ID embedding. This design allows the model to extract global EEG features while explicitly distinguishing between subject-specific distributions, resulting in improved generalization and adaptability to new subjects without retraining an entirely separate model.

With our contributions in this paper, we are focusing on two key issues in the domain of EEG Classification, namely cross-subject training and generalization, as well as regularization for noisy input distribution:

1. We propose LAtte, a fully hyperbolic model for EEG Classification.
2. We enhanced cross-subject learning with subject-specific low-rank adapters.
3. We adapt InceptionTime to learn on a Lorentz manifold in hyperbolic space.
4. We use a Johnson-Lindenstrauss inspired projection for hyperbolic models for improved generalization and computational efficiency.
5. A protocol for fine-tuning models for EEG Classification based on cross-subject training with subject information.
6. We demonstrate the efficacy of LAtte on three well-established datasets in the domain of EEG classification. Here, we achieve an average improvement of 3.41% in the subject-specific and 10.01% in the subject-conditional setting over the state-of-the-art.

## 2 RELATED WORK

**EEG Classification**    In the EEG classification literature, most models adopt a combination of spatial, temporal, or hybrid convolutional layers, typically followed by normalization, pooling, and a final linear classifier. Some pioneering models, such as EEGNet (Lawhern et al., 2018) and ShallowConvNet (Schirrmeister et al., 2017), both employ temporal convolutions within modular convolutional blocks. SCCNet (Wei et al., 2019) extends this design by introducing spatiotemporal convolutions to facilitate spectral feature extraction. Similarly, FBCNet (Mane et al., 2021) adopts an approach aligned with EEG-TCNet but integrates spectral filtering at the input stage.

EEG-TCNet (Ingolfsson et al., 2020) leverages causal convolutions to preserve the temporal structure of the signal, while TCNet-Fusion (Musallam et al., 2021) augments this framework by concatenating intermediate feature maps from the initial layers before classification. MBEEGSE (Altuwaijri et al., 2022) represents one of the earliest applications of transformer architectures to EEG, employing EEG-specific convolutional blocks (Riyad et al., 2020) alongside squeeze-and-excitation attention mechanisms (Altuwaijri & Muhammad, 2022). MAtt (Pan et al., 2022) introduces a novel approach by replacing conventional Euclidean attention layers with manifold attention layers operating in Riemannian space.

The current research on cross-subject training is mostly limited to the task of emotion recognition and not the broader domain of EEG Classification. In Li et al. (2018), the importance of different EEG features is studied, and there are some meta-learning approaches, including contrastive learning Shen et al. (2023), pre-training Cimtay & Ekmekcioglu (2020), and transfer learning Li et al. (2020). More recently, Burchert et al. (2024) studied the cross-subject setting for the task of EEG classification by introducing a training protocol with embedding subject information, addressing cross-subject generalization, and proposing the adaptation of the convolutional baselines

ResNet (Kachuee et al., 2018) and InceptionTime (Ismail Fawaz et al., 2020) from the Time Series Classification domain.

**Hyperbolic** Early developments in hyperbolic deep learning frequently adopted hybrid architectures that paired Euclidean encoders with hyperbolic decoders (Mettes et al., 2024). This design choice mitigated the computational overhead associated with hyperbolic operations and circumvented the absence of well-established hyperbolic analogues for many standard Euclidean components. However, recent work has increasingly shifted towards fully hyperbolic models, motivated by the desire to exploit the geometric inductive biases of hyperbolic space more holistically.

Chen et al. (2022) advanced this direction by introducing hyperbolic counterparts to several foundational components, including fully connected layers, graph convolution layers, and attention mechanisms. Their architecture employs square Lorentzian distance as a similarity measure and learns class prototypes directly in hyperbolic space, using this same metric for classification loss, building on earlier works such as (Atigh et al., 2022), (Kim et al., 2023), and (Mettes et al., 2024).

This approach was later extended to EEG tasks, most notably in the works of Nguyen et al. (2025) and Bdeir et al. (2025). Nguyen et al. (2025) proposed a unified framework for neural networks on symmetric spaces of noncompact type, which include both hyperbolic and SPD manifolds. Their approach is built around a generalized formulation of the point-to-hyperplane distance, from which they derive closed-form expressions to construct hyperbolic fully connected layers and attention mechanisms. Applying this to EEG resulted in great performance gains.

In parallel, Bdeir et al. (2025) introduced HyperMatt, an extension of the Matt framework (Pan et al., 2022) that replaces the original SPD decoder with a Lorentz-based hyperbolic decoder. Specifically, they project the SPD matrices on the Lorentz space and apply Lorentzian attention and a hyperbolic MLR as a final classification layer. Although this represents an important step toward leveraging hyperbolic geometry for EEG, the encoder remains non-hyperbolic. As a result, there may still be possible performance gains from fully capturing the hierarchical relationships inherent in EEG data.

## 3 BACKGROUND

**Lorentz Manifold** Hyperbolic space is a Riemannian manifold characterized by constant negative sectional curvature $-1/K < 0$ where $K$ is the curvature surrogate. In the following work, we adopt the Lorentz model of hyperbolic space (also known as the hyperboloid model), which uses the upper sheet of a two-sheeted hyperboloid within Minkowski space. The $n$-dimensional Lorentz space is then defined as $\mathbb{L}_K^n = (\mathcal{L}^n, \mathfrak{g}_{\boldsymbol{x}})$, where the manifold $\mathcal{L}^n$ is

$$\mathcal{L}^n := \left\{ \boldsymbol{x} = [x_t, \boldsymbol{x}_s] \in \mathbb{R}^{n+1} \,\middle|\, \langle \boldsymbol{x}, \boldsymbol{x} \rangle_{\mathcal{L}} = -K, \ x_t > 0 \right\},$$

The Lorentzian inner product defines the Riemannian metric $\langle \boldsymbol{x}, \boldsymbol{y} \rangle_{\mathcal{L}} := -x_t y_t + \boldsymbol{x}_s^\top \boldsymbol{y}_s$. Following terminology from special relativity, we refer to $x_t$ as the *time* component and $\boldsymbol{x}_s$ as the *space* component. We can then use the definition of the inner product and the terminology to define the origin of the space as $\overline{\boldsymbol{0}}^K = [\sqrt{K}, 0, \dots, 0]^\top$.

**Distance** The shortest path between two points on the manifold follows the curvature of space and is known as a geodesic. For any two points $\boldsymbol{x}, \boldsymbol{y} \in \mathbb{L}_K^n$, the length of the geodesic between them is defined as

$$d_{\mathbb{L}}(\boldsymbol{x}, \boldsymbol{y}) = \sqrt{K} \operatorname{acosh}\left( \frac{-\langle \boldsymbol{x}, \boldsymbol{y} \rangle_{\mathcal{L}}}{K} \right).$$

The formulation for the squared distance, as proposed by Law et al. (2019), is simpler and can be calculated as

$$d_{\mathbb{L}}^2(\boldsymbol{x}, \boldsymbol{y}) = \|\boldsymbol{x} - \boldsymbol{y}\|_{\mathbb{L}}^2 = -2K - 2\langle \boldsymbol{x}, \boldsymbol{y} \rangle_{\mathcal{L}}.$$

**Exponential and Logarithmic Maps.** As a Riemannian manifold, the Lorentz space is locally Euclidean, allowing approximation at any point $\boldsymbol{x}$ using the tangent space $\mathcal{T}_{\boldsymbol{x}}\mathcal{L}$. We present the exponential and logarithmic maps that move points from and to the tangent space in A.1, along with additional Lorentzian operations.

# 4 METHODOLOGY

## 4.1 PROBLEM SETTING

Given a set of EEG sequences from a subject and their classification into distinct categories (e.g., by an expert), the objective is to classify new EEG recordings from this subject into these categories based on patterns in the signals. Let:

- $x^{\text{eeg}} \in X^{\text{eeg}} := \mathbb{R}^{C \times T}$ be an EEG recording (with $C$ channels and $T$ timepoints),
- $x^{\text{id}} \in X^{\text{id}} := \{1, \dots, S\}$ the ID of the subject of this EEG recording and
- $y \in Y := \{1, \dots, K\}$ a class of $(x^{\text{eeg}}, x^{\text{id}})$.

Given $N$ such labeled EEG recordings, i.e., triples $\left((x_1^{\text{eeg}}, x_1^{\text{id}}, y_1), \dots, (x_N^{\text{eeg}}, x_N^{\text{id}}, y_N)\right)$ from an unknown distribution $d$ (e.g., representing a subject), the task is to find a model $\hat{y}$ that maps EEG signals $X^{\text{eeg}}$ to the correct class (where $X^{\text{eeg}}$ and its ground truth class $y$ are from the same distribution $d$).

## 4.2 CROSS-SUBJECT TRAINING WITH LORA

Conventional EEG classification methods typically train separate models for each subject, motivated by the assumption that inter-subject variability in neural patterns and data distributions hinders effective joint training. However, this subject-specific paradigm suffers from poor generalization: models specialized on individual subjects often fail to perform reliably on unseen subjects, limiting their applicability in real-world clinical settings where diagnostic predictions are typically made once per subject. This motivates the development of models that generalize well across individuals. Previous work embeds the subject information and concatenates it with the channel dimension of the input data to the model, making the subject information available to the model (Burchert et al., 2024). However, this has several disadvantages, especially in convolutional networks, due to the position of the encoding and mixing it with the EEG channel information. We model a richer embedding of the subject information by adding low-rank adapters (Hu et al., 2022). Let

$$\hat{y} : X^{\text{eeg}} \times X^{\text{id}} \to Y$$

be an EEG classification model, parametrized by $P$ many parameter matrices $W_p \in \mathbb{R}^{I_p \times J_p}$ ($p = 1{:}P$). We add low-rank adapter modules to the parameter matrices $W_p$, replacing the original $W_p$, now called $W_p^{\text{shared}}$, by a combination of shared and subject-specific parameters:

$$W_p := W_p^{\text{shared}} + Q_p^s (R_p^s)^T, \quad W_p^{\text{shared}} \in \mathbb{R}^{I_p \times J_p}, \quad Q_p^s \in \mathbb{R}^{I_p \times K_p}, R_p^s \in \mathbb{R}^{J_p \times K_p} \tag{1}$$

with a lower rank $K_p \ll I_p, J_p$ for all $p$.

The model now is learned as before, it just has more parameter matrices and more parameters. From initially $\sum_p I_p J_p$ parameters it grows to

$$\sum_p I_p J_p + (I_p + J_p) K_p S$$

We initialize $Q_p^s$ with a random Gaussian and $R_p^s$ with zeros. Thus, $Q_p^s R_p^{s^T}$ is zero at the beginning of training. This enables a rich and lightweight integration of the subject information in the training process because the parameters of the model $p$ are directly dependent on the subject $s$.

## 4.3 LATTE: LORENTZ ATTENTION FOR EEG CLASSIFICATION

**Processor**  Our model LAtte utilizes the EEG sequence $X^{\text{eeg}}$ as well as subject information $X^{\text{id}}$. For the initial processing of the data, we apply Spatial Component Analysis (SCA) followed by a Spatio-Temporal Filtering (STF) inspired by Wei et al. (2019). It consists of two convolutional blocks followed by a batchnorm. The first block mimics a spatial component analysis that decomposes the original EEG data from the channel domain to a component domain using a kernel size of $(F_T, 1)$, which produces a linear combination over all channels. The second convolution is of size $(1, F_C)$ convolving over the time dimension and extracting temporal features. Additionally, we

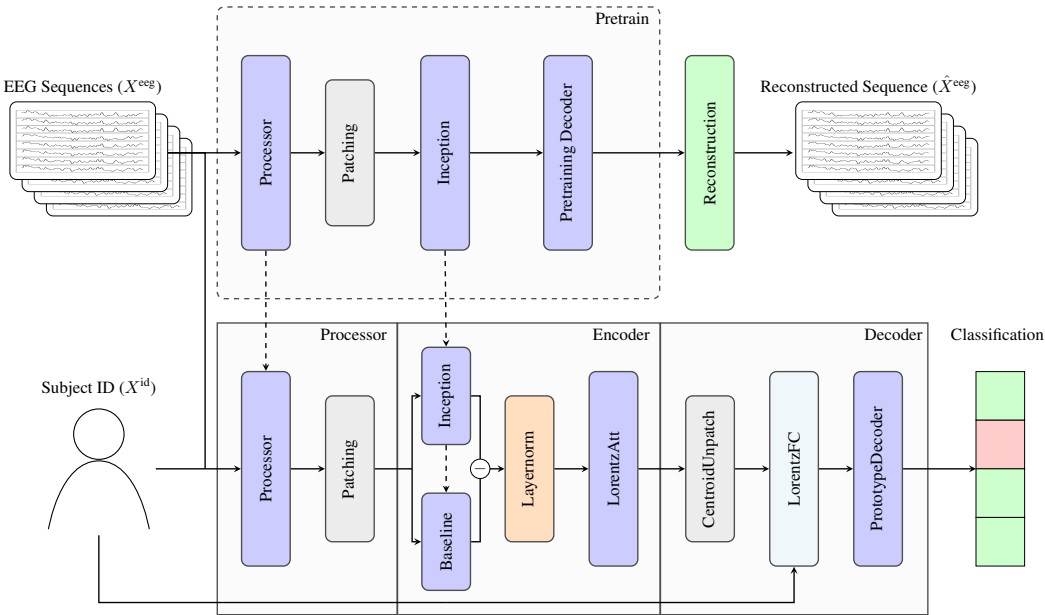

Figure 1: LAtte architecture overview.

inject the subject information by following Equation (1) so that the parameter matrix expands to $W_{\text{proc}} := W_{\text{proc}}^{\text{shared}} + Q_{\text{proc}}^s (R_{\text{proc}}^s)^T$ for both SCA and STF respectively.

$$Z = \text{BN}\left(\text{STF}\left(\text{BN}\left(\text{SCA}\left(X^{\text{eeg}}, X^{\text{id}}\right)\right)\right)\right) \qquad (2)$$

After creating the initial embeddings of the data, we map the latent representation onto the Lorentz manifold in hyperbolic space. Next, we patch the sequence into non-overlapping segments to capture local temporal patterns more effectively. After embedding the input, we can use the centroid to take the Lorentzian average of the patch embeddings. This approach improves the representation learning of the model. Additionally, it allows efficient processing, especially in Transformer-based architectures, by reducing the sequence length.

**Encoder** Previous studies in EEG analysis have often leveraged paired baseline-task recordings, where a baseline graph is constructed from resting-state and a task graph is derived from signals recorded during the task Maess et al. (2016). Subtracting the baseline graph from the task graph helps isolate task-specific neural activity by mitigating individual variability and non-task-related noise. However, in many real-world EEG datasets, explicit baseline recordings are not available, which poses a challenge for applying this method to model clean task-related representations.

To address this limitation, we propose a dual-branch architecture that learns an implicit baseline representation directly from task data. Specifically, we introduce a Hyperbolic Inception Block with average pooling to estimate an average baseline graph and a standard Inception Block with max pooling to emphasize task-specific activations and construct a task graph. Finally, we subtract the learned baseline graph from the task graph as can be seen in Figure 1 and Appendix A.2, which should result in a cleaned representation that better highlights task-relevant neural activity.

**Hyperbolic InceptionTime** To effectively model latent representations across multiple temporal resolutions and extract meaningful features, we utilize the InceptionTime architecture Ismail Fawaz et al. (2020). Specifically, we translate the original Euclidean architecture to the Lorentz manifold to capture cross-channel and cross-temporal hierarchical relationships.

The block begins with a bottleneck layer to reduce the channel dimensionality, followed by three parallel convolutional branches with different kernel sizes to capture temporal patterns at multiple

resolutions. In parallel, a max-pooling branch is included to emphasize prominent local features. For this, we use our formulation of a Lorentzian maxpool defined in the section below. For the task block, we replace the max-pooling branch with an average-pooling branch by using the centroid-based average pooling proposed in Bdeir et al. (2024).

We then summarize the block as follows. Given input embeddings $Z \in \mathcal{L}_K^n$

$$Z_{\mathrm{b}} = \mathrm{ReLU}\left(\mathrm{BN}\left(\mathrm{Conv1D}(Z,\ W_{\mathrm{b}})\right)\right), \quad W_{\mathrm{b}} \in \mathbb{R}^{C' \times C \times 1} \tag{3}$$

where $C' < C$ is the bottleneck dimension. The convolutions are applied several times with different kernel sizes in parallel:

$$Z_i = \mathrm{ReLU}\left(\mathrm{BN}\left(\mathrm{Conv1D}(Z_{\mathrm{b}}, W_i)\right)\right), \quad W_i \in \mathbb{R}^{C'' \times C' \times k_i} \tag{4}$$

for each $k_i \in \{k_1, k_2, k_3\}$, where $C''$ is the number of filters per path. The outputs of all the branches are then concatenated using direct Lorentz Concatenation (HCat) Qu & Zou. :

$$\mathrm{HCat}(\{x_i\}_{i=1}^N) = \left[\ \sqrt{\sum_{i=1}^4 Z_{i,t}^2 + \frac{N-1}{K}},\ Z_{1,v},\ Z_{2,v},\ Z_{3,v},\ Z_{4,v}\ \right] \tag{5}$$

where $Z_4, v$ is the output of pooling convolution, $K$ is the manifold curvature surrogate, and $t, v$ refer to the time and space values of the vector.

**Hyperbolic MaxPool** We introduce a distance-based Lorentzian max-pooling operator. Given a 1D window of width $k$ (kernel size), stride $t$, padding $u$, and dilation $\delta$, we first compute a scalar score for each embedding given by the geodesic distance to the origin $o$:

$$s_{b,\ell} = d_{\mathcal{L}}(x_{b,\ell}, o) = \mathrm{arcosh}\left(-\langle x_{b,\ell}, o\rangle_L\right), \tag{6}$$

We then perform ordinary maxpool on the scalar score sequence $\{s_{b,\ell}\}_{\ell=1}^L$ to obtain pooled scores and their respective indices. As such, instead of pooling the coordinate value directly, we use these indices to gather the corresponding hyperbolic vectors from the original sequence, which gives the final output

$$Y_{b,j} = x_{b,\ \arg\max_{\ell \in \mathcal{W}_j} s_{b,\ell}}, \tag{7}$$

where $\mathcal{W}_j = \{\ell_0 + \delta \cdot m \mid m = 0, \ldots, k-1\}$, $\ell_0$ is the start of window $j$ determined by $(t, p)$. Because $Y_{b,j}$ is selected from $\{x_{b,\ell}\}$, it remains on $\mathcal{L}^d$ without projection or retraction. Intuitively, this performs geodesic-radius pooling, where within each window we retain the point furthest from the hyperbolic origin, a choice that correlates with the highest semantic difference in hierarchical embeddings. However, due

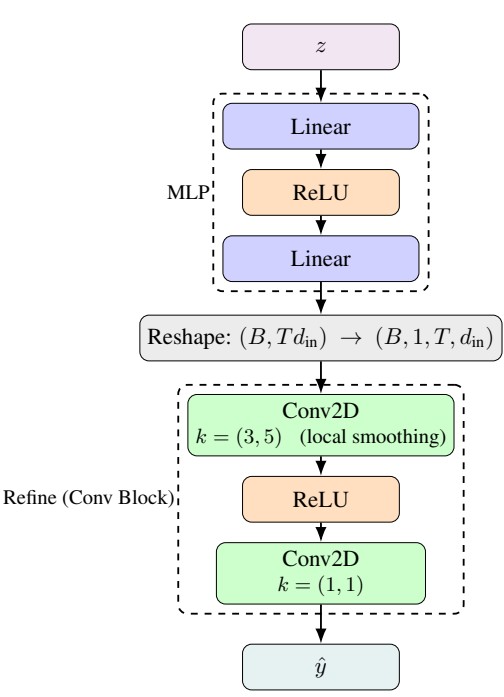

Figure 2: Euclidean decoder used in the reconstruction and cutfill pretraining

to the possible instability of operating on vectors too close to the representational radius of the Lorentz space Mishne et al. (2023), we employ maximum distance rescaling Bdeir et al. (2025).

**Lorentz Fully Connected Layer** After the Lorentz Attention, we apply an LFC to reduce the dimensionality and prepare the embedding for the final classification output. During the training, the $W_{\mathrm{LorentzFC}}^{\mathrm{shared}}$ are frozen to simulate a random projection (Blum, 2005). The assumption here is that given a set of points in a high-dimensional space, the random projection of these points allows us to embed them into a space of much lower dimension in such a way that distances between points are approximately preserved. Let $0 < \varepsilon < 1$ and $X \subset \mathbb{R}^d$ be a set of $n$ points. There exists a linear map

$$f : \mathbb{R}^d \to \mathbb{R}^k, \quad k = O\left(\frac{\log n}{\varepsilon^2}\right),$$

By freezing the LFC with randomly initialized weights $W_{\text{LFC}}^{\text{shared}} \sim \mathcal{U}(-\sigma, \sigma)$, we are projecting the embeddings on the manifold to a lower dimension, reducing computational cost and adding regularization. To compensate for the frozen weights, we add a LoRA again to learn subject-specific features and improve the encoding for the final classification, where $Q_{\text{LFC}}$ follows the same initialization as $W_{\text{LFC}}^{\text{shared}}$ and $R_{\text{LFC}}$ is set to zero. In addition, we set a learning rate $\alpha_s$. Thus, the output is defined as

$$Z = W_{\text{LFC}}^{\text{shared}} Z + \alpha_s Q_{\text{LFC}} R_{\text{LFC}}^T Z \tag{8}$$

## 4.4 PRETRAINING

Our architecture relies on the model's ability to provide a thorough representation of the input data. However, due to the limited amount of data per subject in smaller datasets, we often suffer from overfitting issues that focus on the subject-specific noise. To mitigate and extract meaningful patterns from the EEG signals, we propose the use of a multi-subject encoder pre-training.

The setting is self-supervised, which allows us to train the model without expert labels. We rely on two main surrogate tasks: cut and fill, and reconstruction.

**Cut and Fill** In this task, from the input data, we randomly select an interval of length $l$ uniformly sampled from a range $[l_{min}, l_{max}]$. We then replace the values of the interval with the mean of the input data and predict the intermediate values. The cut-fill can be seen in Algorithm 1. The loss for a given cut-mask $M$ is:

$$\mathcal{L}_{\text{cf}} = \frac{1}{\|M\|_1} \left\| M \odot (\hat{x} - x) \right\|_F^2, \tag{9}$$

---

**Algorithm 1** Apply-Cut-and-Fill

---

**Require:** $x \in \mathbb{R}^{B \times 1 \times C \times T}$
**Require:** $l_{\min}, l_{\max} \in (0, 1]$
**Require:** $f \in \mathbb{R}^{1 \times 1 \times C \times 1}$
**Ensure:** $x^{\text{masked}} \in \mathbb{R}^{B \times 1 \times C \times T}$, $M \in \{0, 1\}^{B \times 1 \times C \times T}$, spans $\{(s_b, e_b)\}_{b=1}^B$
1: $x^{\text{masked}} \leftarrow x; \quad M \leftarrow \mathbf{0}_{B \times 1 \times C \times T}$
2: **for** $b = 1$ **to** $B$ **do**
3: $\quad \ell_b \sim \text{Unif}\left(\{\lfloor l_{\min} T \rfloor, \ldots, \lfloor l_{\max} T \rfloor\}\right)$
4: $\quad s_b \sim \text{Unif}\left(\{0, \ldots, T - \ell_b\}\right); \quad e_b \leftarrow s_b + \ell_b$
5: $\quad$ **for** $c = 1$ **to** $C$ **do**
6: $\quad\quad$ **for** $t = s_b$ **to** $e_b - 1$ **do**
7: $\quad\quad\quad x_{b,1,c,t}^{\text{masked}} \leftarrow f_{1,1,c,1}$
8: $\quad\quad\quad M_{b,1,c,t} \leftarrow 1$
9: $\quad\quad$ **end for**
10: $\quad$ **end for**
11: **end for**
12:
13: **return** $x^{\text{masked}}, M, \{(s_b, e_b)\}_{b=1}^B$

---

**Reconstruction** The reconstruction task is a more conventional self-supervised objective where the EEG data is first passed through the encoder to generate a compact latent representation. This embedding is then fed into the decoder, which attempts to reconstruct the original input signal as closely as possible.

In each pre-training step we stochastically choose which surrogate task to optimize: with probability $r$, we perform a reconstruction update (minimizing $\mathcal{L}_{\text{recon}}$), and with probability $1 - r$ we perform a cut-and-fill update (minimizing $\mathcal{L}_{\text{cf}}$).

**Pretraining Decoder** To facilitate the pertaining process, we rely on a relatively simple decoder composed of a two-layer MLP followed by a convolution block to refine the reshaped outputs. Thus,

$$\hat{y} = \text{FC}_3(a(\text{Conv}_1(\text{FC}_2(a(\text{FC}_1(z))))))$$

where $a$ is the ReLU activation function, FC is a fully connected layer, and Conv is a 2d convolutional layer. This is presented in Figure 2.

## 4.5 REGULARIZATION

To minimize overfitting, we combine fast Johnson–Lindenstrauss (JL) random projections with an EEG cut-and-fill augmentation. The JL layer maps each window to a lower-dimensional subspace using a fixed, sparse random matrix (no trainable parameters), preserving distances while reducing model capacity and subject-specific noise. Additionally, we use a cut-only variant of the previous pretraining task as a lightweight augmentation.

Table 1: Performance comparison for the subject-specific task on the EEG datasets MI, SSVEP, and ERN. We repeated the experiments 10 times on random seeds and report the average accuracy for MI and SSVEP and the AUC for ERN. The best result is highlighted in bold, the second best underlined. The first block contains subject-specific (SS) models, where one model is trained per subject. In the second block are subject-conditional (SC) models, where the model is trained across subjects with additional subject information.

| Models | MI | SSVEP | ERN |
|---|---|---|---|
| ShallowConvNet | 61.84±6.39 | 56.93±6.97 | 71.86±2.64 |
| EEGNet | 57.43±6.25 | 53.72±7.23 | 74.28±2.47 |
| SCCNet | 71.95±5.05 | 62.11±7.70 | 70.93±2.31 |
| EEG-TCNet | 67.09±4.66 | 55.45±7.66 | 77.05±2.46 |
| TCNet-Fusion | 56.52±3.07 | 45.00±6.45 | 70.46±2.94 |
| FBCNet | 71.45±4.45 | 53.09±5.67 | 60.47±3.06 |
| MBEEGSE | 64.58±6.07 | 56.45±7.27 | 75.46±2.34 |
| InceptionTime | 62.85±3.21 | 62.71±2.95 | 73.55±5.08 |
| MAtt | **74.71**±5.01 | 65.50±8.20 | 76.01±2.28 |
| HyperMAtt | 74.12±2.91 | 68.10±2.41 | 78.01±1.30 |
| LAtte | 73.77±3.11 | **73.47**±1.82 | **80.55**±2.22 |
| ResNetJoint | 55.54±2.72 | 54.15±1.19 | 73.09±0.72 |
| MAttJoint | 61.13±0.56 | 60.71±0.29 | 75.78±1.23 |
| InceptionJoint | 61.38±1.57 | 66.00±0.36 | 76.13±0.95 |
| LAtteJoint | **71.63**±2.80 | **71.53**±1.00 | 79.85±2.12 |
| SS: Relative Δ | -1.25% | 7.89% | 3.26% |
| SC: Relative Δ | 16.67% | 8.38% | 4.98% |

## 4.6 SUBJECT-CONDITIONAL FINE-TUNING

Despite challenges such as subject-specific noise and distributional shifts, learning shared structure across subjects remains a promising strategy for constructing more robust latent representations. Building on prior work of fine-tuning for EEG Classification (Wei et al., 2019) and cross-subject training with subject features (Burchert et al., 2024), we combine both approaches and propose a fine-tuning protocol with subject features. First, we train the cross-subject model with the LoRA embedding of the subject information and then fine-tune on the individual subject, leveraging universal patterns shared across subjects while making the different distributions explicit to the model.

## 5 EXPERIMENTS

**Experimental Setup** We evaluate our approach on three widely studied EEG datasets, each representing a classification paradigm: motor imagery, steady-state visual responses, and error-related brain activity. For the preprocessing, we follow the common steps and use the same train/val/test splitting protocol as Pan et al. (2022), where for the tasks of SSVEP and ERN, the data is split session-wise, and for MI, instance-wise. A detailed description can be found in Appendix A.3.

We compare our model LAtte against well-established EEG Classification baselines, including ShallowConvNet (Schirrmeister et al., 2017), EEGNet (Lawhern et al., 2018), SCCNet (Wei et al., 2019), EEG-TCNet (Ingolfsson et al., 2020), TCNet-Fusion (Musallam et al., 2021), FBCNet (Mane et al., 2021), MBEEGSE (Altuwaijri et al., 2022), InceptionTime (Burchert et al., 2024), MAtt (Pan et al., 2022), and HyperMatt (Bdeir et al., 2025) across three commonly used datasets. Furthermore, we include three cross-subject baselines: ResNetJoint, MAttJoint, and InceptionJoint (Burchert et al., 2024). For the MI and SSVEP datasets, accuracy is used as the performance metric, while AUC is employed for ERN due to a class imbalance. For LAtte, we first pre-train the model with the reconstruction task and load the resulting weights for the processor and LorentzInceptionTime layers. In the main training loop, LAtte is trained on all subjects jointly.
Hyperparameters were selected based on validation accuracy and AUC for the ERN task. The search ranges and optimal configurations for each dataset are reported in Appendix A.4. LAtte is trained for 300 epochs, and test performance is reported at the epoch with the best validation score. Experiments are repeated 10 times with different random seeds.

**Performance Comparison** In Table 1, we compare our proposed model against current state-of-the-art baselines, where baseline performance is aggregated from Pan et al. (2022); Burchert et al. (2024); Bdeir et al. (2025). The first block reports results in the subject-specific (SS) setting, where a separate model is trained for each subject, and the final performance is averaged across subjects. The second block presents results in the subject-conditional (SC) setting, where a single model is trained across all subjects with additional subject meta-features (e.g., subject ID). This design enables generalization to unseen subjects without retraining, which is particularly important for clinical applications.

In the SC setting, LAtteJoint achieves a significant performance improvement over all baselines across tasks. Unlike existing models that rely on static subject embeddings, LAtteJoint leverages a richer LoRA-based subject representation. This more expressive embedding enables better separation of subject-specific noise distributions. In addition, LAtte outperforms the subject-specific baselines on SSVEP and ERN tasks, highlighting the benefits of its more expressive hyperbolic representation.

For the SS setting, we further fine-tune our cross-subject model on individual subjects using only their instances and subject ID, without additional training data. We report the mean performance across all subjects and the pooled standard deviation over runs. The per-subject results are provided in Appendix A.4. Fine-tuning improves the cross-subject model by allowing it to capture the subject's distribution directly, while still benefiting from universal patterns learned by the joint model. Moreover, training converges within a few iterations due to the strong initialization and the fully subject-dependent LoRA pre-decoder.

**Ablation Studies** In Table 2, we analyze the contribution of each component in the LAtte architecture by removing one module at a time. The largest performance drop occurs when replacing the Lorentz manifold with Euclidean space. Here, accuracy decreases by 19.75%, confirming the importance of a more expressive geometric representation. Removing the subject-dependent LoRA parameters also leads to a substantial drop by 12.11%, as the model can no longer effectively distinguish subject-specific distributions, which is critical for cross-subject generalization. The decoder projection and cut/fill augmentation contribute to improved regularization and stable convergence, which is an essential property for small EEG datasets. Finally, pretraining the processor and encoder provides additional gains by offering more stable initialization.

Table 2: Component Analysis of LAtte. We compare the performance of the full LAtte model to versions without the main modules on SSVEP. Here, we report the accuracy and standard deviation over 5 runs.

| Model Variant | Accuracy | $\Delta$ |
|---|---|---|
| LAtte Full | 71.53±1.00 | - |
| LAtte w/o Lorentz | 57.40±6.53 | -19.75% |
| LAtte w/o LoRA | 62.87±0.51 | -12.11% |
| LAtte w/o Proj | 69.34±1.37 | -3.06% |
| LAtte w/o Cut/Fill | 69.55±0.71 | -2.77% |
| LAtte w/o Pretrain | 69.71±0.68 | -2.54% |

## 6 CONCLUSION

In this work, we introduced LAtte, a novel hyperbolic framework for cross-subject EEG classification that unifies Lorentzian attention with an InceptionTime-based encoder. By combining subject-specific low-rank adapters with hyperbolic representations, LAtte effectively learns shared neural patterns from subject-dependent variability. This design enables a single unified model to generalize across individuals while maintaining adaptability to new, unseen subjects. This is critical in clinical applications, and the proposed cross-subject outperforms most of its subject-specific counterparts. Our results on three well-established datasets highlight two broader messages: first, that hyperbolic geometry offers a natural and powerful inductive bias for modeling the hierarchical structure inherent in EEG data; and second, that subject-adaptive mechanisms such as LoRA can enable scalable, generalizable EEG decoding. Taken together, LAtte points toward a path where EEG classification is both more robust to noise and more accessible in real-world BCI applications.

# 7 REPRODUCIBILITY STATEMENT

We are committed to ensuring the reproducibility of our model, LAtte. To support this, we provide the source code, including pre-processing steps, model architecture, and training scripts, in the supplementary material. Upon publication, the code will also be made publicly available on GitHub, along with detailed documentation to guide experiment setup and execution. Additionally, we commit to releasing the pretrained checkpoints used for writing this paper. The datasets used in our experiments are publicly available, and we additionally provide a link to the preprocessed data in a cloud service for easy access. All hyperparameters and model configurations are detailed in both the paper and the code repository to ensure easy replication.

# 8 ETHICS STATEMENT

We are committed to contributing positively to society and human well-being, while respecting the privacy of the subjects involved in our evaluation. Our experiments utilize three established EEG datasets: MI, SSVEP, and ERN, which involve data from human subjects. These datasets are fully anonymized and do not contain any personally identifiable information. Additionally, they are publicly available and widely used within the machine learning community.

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

# A  APPENDIX

## A.1  ADDITIONAL HYPERBOLIC OPERATIONS

**Exp and Log Maps**    The exponential map $\mathcal{T}_{\boldsymbol{x}} \to \mathbb{L}_K^n$ projects tangent vectors onto the manifold via:

$$\exp_{\boldsymbol{x}}^K(\boldsymbol{z}) = \cosh(\alpha)\boldsymbol{x} + \sinh(\alpha)\frac{\boldsymbol{z}}{\alpha}, \quad \alpha = \sqrt{1/K}\, \|\boldsymbol{z}\|_{\mathbb{L}},$$

where $\|\boldsymbol{z}\|_{\mathbb{L}} = \sqrt{\langle \boldsymbol{z}, \boldsymbol{z}\rangle_{\mathcal{L}}}$. The inverse operation, the logarithmic map, is given by:

$$\log_{\boldsymbol{x}}^K(\boldsymbol{y}) = \frac{\mathrm{acosh}(\beta)}{\sqrt{\beta^2 - 1}}(\boldsymbol{y} - \beta\boldsymbol{x}), \quad \beta = -\frac{1}{K}\langle \boldsymbol{x}, \boldsymbol{y}\rangle_{\mathcal{L}}.$$

**Lorentzian Centroid**    (Law et al., 2019) derive a closed-form for a Lorentzian centroid $\boldsymbol{\mu}_{\mathbb{L}}$ based on the square distances. For points $\{\mathbf{x}_i\}_{i=1}^m \subset \mathbb{L}_K^n$ with weights $\boldsymbol{\nu} \in \mathbb{R}^m$, it is computed as:

$$\boldsymbol{\mu}_{\mathbb{L}} = \frac{\sqrt{K} \cdot \sum_{i=1}^m \nu_i \mathbf{x}_i}{\left\| \sum_{i=1}^m \nu_i \mathbf{x}_i \right\|_{\mathcal{L}}}, \tag{10}$$

where $\|\mathbf{z}\|_{\mathcal{L}} := \sqrt{|\langle \mathbf{z}, \mathbf{z}\rangle_{\mathcal{L}}|}$ ensures normalization to the hyperboloid surface. This closed-form solution approximates the centroid while maintaining manifold constraints.

**Lorentz Prototype Decoder**    A strength of hyperbolic machine learning approaches is the ability to produce fine embeddings for input data while minimally distorting the hierarchical relationships between them. This leads to a better use of the ambient embedding space and better instance clustering. To leverage this, we utilize prototypical classification heads. As opposed to the MLR classifier proposed in Bdeir et al. (2024), prototype classifiers employ class embeddings as cluster points and outputs the results as distances to each cluster midpoint. The center with minimal distance is the the most similar semantically and is used as the classified class. The Lorentzian prototype decoder in this work is the direct translation of the decoder proposed by Wang et al. (2021). We first randomly initialize N points on the hyperboloid using the Wrapped Normal Lorentz distribution in Nagano et al. (2019). We then measure similarity using the square distance as it is better defined and computationally more efficient.

Additionally, this layer can be set to non-learnable, which forces the class centroids to remain static during the training process. This leads to better generalization in some datasets and avoids overfitting. We use cross-entropy as the objective function. Thus,

$$\ell(X^{\mathrm{eeg}}, X^{\mathrm{id}}, y; W; \mathcal{L}) := \frac{1}{N}\sum_{n=1}^N \mathrm{cross\text{-}entropy}(y_n, \hat{y}_n) \quad \text{with } \hat{y} := \mathrm{LAtte}(X^{\mathrm{eeg}}, X^{\mathrm{id}}; W; \mathcal{L}) \tag{11}$$

## A.2 LATTE

In Algorithm 2, we show the LAtte model modules with their respective inputs and parameters. Additionally, we have assigned each block the corresponding Equation for easy cross-reference.

**Lorentz Fully Connected Layer**   After the Lorentz Attention, we apply an LFC to reduce the dimensionality and prepare the embedding for the final classification output. During the training, the $W_{\text{LorentzFC}}^{\text{shared}}$ are frozen to simulate a random projection following the Johnson-Lindenstrauss lemma (Johnson et al., 1984), which states that a set of points in a high-dimensional space can be embedded into a space of much lower dimension in such a way that distances between points are preserved. Let $0 < \varepsilon < 1$ and $X \subset \mathbb{R}^d$ be a set of $n$ points. There exists a linear map

$$f : \mathbb{R}^d \to \mathbb{R}^k, \quad k = O\left(\frac{\log n}{\varepsilon^2}\right),$$

By freezing the LFC with randomly initialized weights $W_{\text{LFC}}^{\text{shared}} \sim \mathcal{U}(-\sigma, \sigma)$, we are projecting the embeddings on the manifold to a lower dimension, reducing computational cost and adding regularization. To compensate for the frozen weights, we add a LoRA again to learn subject-specific features and improve the encoding for the final classification, where $Q_{\text{LFC}}$ follows the same initialization as $W_{\text{LFC}}^{\text{shared}}$ and $R_{\text{LFC}}$ is set to zero. In addition, we set a learning rate $\alpha_s$. Thus, the output is defined as

$$Z = W_{\text{LFC}}^{\text{shared}} Z + \alpha_s Q_{\text{LFC}} R_{\text{LFC}}^T Z \tag{12}$$

**Lorentz Multi-Head Attention**   We compute Lorentzian queries, keys, and values with Lorentz Fully Connected layers (LFC) on the manifold:

$$Q = \text{LFC}_q(X), \quad K = \text{LFC}_k(X), \quad V = \text{LFC}_v(X).$$

The attention weights are obtained from scaled Lorentzian squared distances

$$\alpha_{ij} = \frac{\exp\left(-\frac{\lambda}{\tau} d_{\mathcal{L}}^2(Q_i, K_j)\right)}{\sum_{j'} \exp\left(-\frac{\lambda}{\tau} d_{\mathcal{L}}^2(Q_i, K_{j'})\right)},$$

where $\lambda$ is a learnable scaling factor and $\tau$ is the temperature. Values are aggregated by the Lorentzian weighted centroid

$$\widetilde{V}_i = \text{Centroid}_{\mathcal{L}}(V_j, \alpha_{ij}),$$

and the final output is given by

$$\text{LorentzAtt}(X) = \text{LFC}_o(\widetilde{V}), \tag{13}$$

with $\text{LFC}_o$ a Lorentz fully connected layer.

## A.3 DATASETS

**MI — Motor Imagery** (Brunner et al., 2008). The MI dataset, originally released as BCI Competition IV-2a (2008), is a cornerstone benchmark for motor imagery classification. It contains EEG recordings from 9 subjects, acquired with 22 Ag/AgCl electrodes over central and surrounding scalp regions at a sampling rate of 250 Hz. The experimental task involves four motor imagery conditions: right hand, left hand, feet, and tongue. Following established preprocessing protocols, signals were down-sampled to 128 Hz, band-pass filtered to 4–38 Hz, and segmented into 4-second epochs starting 0.5 s post-cue, yielding 438 time points per trial across 22 channels.

**SSVEP — Steady-State Visual Evoked Potentials** (Nikolopoulos, 2021). The SSVEP dataset (MAMEM II, 2016) targets frequency-tagged visual responses. EEG was collected from 11 subjects using the EGI 300 Geodesic EEG System. Participants fixated on one of five flickering visual stimuli (6.66, 7.50, 8.57, 10.00, 12.00 Hz) for 5-second intervals. Preprocessing retained 1–50 Hz activity and focused on 8 occipital channels (PO7, PO3, POz, PO4, PO8, O1, Oz, O2), corresponding to the visual cortex. Each trial was partitioned into four non-overlapping 1-second segments (125 samples per channel), producing  500 trials per subject of 8-channel SSVEP data.

---

**Algorithm 2** LAtte: Lorentz Attention for EEG Classification

---

**Require:** Training dataset $\mathcal{D}^{\text{train}} = \{(x^{\text{eeg}}, x^{\text{id}}, y_1), \ldots, (x_N^{\text{eeg}}, x_N^{\text{id}}, y_N)\}$, $\mathcal{L}^n$, $W_{\text{proc}}^{\text{pretrained}}$, $W_{\text{inc}}^{\text{pretrained}}$, number of epochs $M$

 Initialization :$W_{\text{proc}}^{\text{shared}} \leftarrow W_{\text{proc}}^{\text{pretrained}}$

$$W_{\text{incMax}}, W_{\text{incAvg}} \leftarrow W_{\text{inc}}^{\text{pretrained}}$$
$$W_{\text{att}}, W_{\text{dec}}^{\text{shared}}, W_{\text{out}}, Q_{\text{dec}} \sim \mathcal{U}(-\sigma, \sigma)$$
$$Q_{\text{proc}} \sim \mathcal{N}(\mu, \sigma)$$
$$R_{\text{proc}}, R_{\text{dec}} = 0$$

 **for** epoch $m = 1$ to $M$ **do**
  **for** each training example $(x_n^{\text{eeg}}, x_n^{\text{id}}, y_n) \in \mathcal{D}^{\text{train}}$ **do**
   $Z_n \leftarrow \text{Processor}(x_n^{\text{eeg}}, x^{\text{id}}; W_{\text{proc}}^{\text{shared}}; Q_{\text{proc}}, R_{\text{proc}})$  {*Equation* (2)}
   $Z_n \leftarrow \text{ToHyperbolic}(Z_n; \mathcal{L}^n)$
   $Z_n \leftarrow \text{Patching}(Z_n)$
   $Z_{base} \leftarrow \text{BaselineBlock}(Z_n; W_{\text{incMax}})$  {*Equation* (5)}
   $Z_{inc} \leftarrow \text{InceptionBlock}(Z_n; W_{\text{incAvg}})$  {*Equation* (5)}
   $Z_n \leftarrow Z_{base} - Z_{inc}$
   $Z_n \leftarrow \text{Layernorm}(Z_n)$
   $Z_n \leftarrow \text{LorentzAttention}(Z_n; W_{\text{att}})$  {*Equation* (13)}
   $Z_n \leftarrow \text{Centroid Unpatch}(Z_n)$
   $Z_n \leftarrow \text{LorentzFC}(Z_n, x^{\text{id}}; W_{\text{dec}}^{\text{shared}}; Q_{\text{dec}}, R_{\text{dec}})$  {*Equation* (12)}
   $\hat{y}_n \leftarrow \text{LorentzPrototypeDecoder}(Z_n; W_{\text{out}})$  {*Equation* (12)}
   $\ell_n \leftarrow \text{cross-entropy}(\hat{y}_n, y_n)$  {*Equation* (11)}
  **end for**
  Compute total loss: $\ell \leftarrow \frac{1}{N} \sum_{n=1}^{N} \ell_n$
  Update parameters: $W, W^{\text{shared}}, Q_p^s, R_n^p$ based on $\nabla_\theta \mathcal{L}$
 **end for**
 **return** $W$

---

**ERN — Error-Related Negativity** (Margaux et al., 2012). The ERN dataset, released as part of the 2015 BCI Challenge[1], captures error-related brain responses during a P300-based spelling task. EEG was recorded from 16 subjects using 56 Ag/AgCl electrodes at 600 Hz. The task poses a binary classification problem, with a natural imbalance favoring correct responses. Preprocessing included down-sampling to 128 Hz and band-pass filtering between 1–40 Hz. Each trial was represented by 56 channels with 160 time points, offering a challenging benchmark due to its class imbalance and inter-subject variability.

## A.4 EXPERIMENTS

In Appendix A.4, we show the performance of our fine-tuned model LAtte and compare it to recent baselines. Here, the EEG common subject variability can be observed, where subject 3 defaults to near random performance while subject 2 achieves almost perfect accuracy. This trend is persistent for all models. However, on previously challenging subjects, e.g., 4 and 5, LAtte, with its pretraining and richer subject information, is capable of leveraging features that are shared across the dataset to increase performance.

The hyperparameter search for the learning rate $\{1e^{-3}, 1e^{-4}\}$, weight decay, $\{1e^{-1}, 1e^{-2}, 5e^{-2}, 1e^{-3}\}$, patching windows $\{1,2,3,4\}$ and learning rate of the LoRA weights in the decoder $\{1e^{-1}, 1e^{-3}, 1e^{-5}\}$ was repeated on random seeds and the optimal values selected on the best validation accuracy for MI and SSVEP and the best validation AUC for ERN. The optimal values are shown in Table 4.

---

[1] https://www.kaggle.com/c/inria-bci-challenge

Table 3: Performance Comparison on the SSVEP dataset for the subject-specific case. We report the average accuracy over 10 runs, respectively.

| Subject | InceptionTime | MAtt | HyperMAtt | LAtte |
|---------|--------------|------|-----------|-------|
| 1 | $80.40 \pm 2.06$ | $81.60 \pm 2.87$ | $\underline{89.94} \pm 1.29$ | $\mathbf{90.00} \pm 2.18$ |
| 2 | $86.60 \pm 1.62$ | $89.40 \pm 1.36$ | $\underline{90.31} \pm 0.96$ | $\mathbf{94.50} \pm 4.62$ |
| 3 | $61.60 \pm 3.07$ | $58.20 \pm 5.64$ | $\underline{68.05} \pm 3.06$ | $\mathbf{74.20} \pm 0.16$ |
| 4 | $25.00 \pm 4.00$ | $20.60 \pm 3.88$ | $\underline{30.00} \pm 0.58$ | $\mathbf{42.67} \pm 1.89$ |
| 5 | $25.00 \pm 6.72$ | $26.40 \pm 4.80$ | $\underline{30.96} \pm 1.53$ | $\mathbf{53.20} \pm 2.34$ |
| 6 | $79.20 \pm 1.72$ | $79.00 \pm 2.68$ | $\underline{80.90} \pm 1.55$ | $\mathbf{86.40} \pm 0.70$ |
| 7 | $69.20 \pm 1.72$ | $66.00 \pm 2.19$ | $\underline{69.08} \pm 2.08$ | $\mathbf{78.60} \pm 2.07$ |
| 8 | $23.60 \pm 1.74$ | $23.80 \pm 2.71$ | $\mathbf{29.04} \pm 0.58$ | $\underline{25.20} \pm 2.03$ |
| 9 | $79.40 \pm 2.58$ | $88.20 \pm 2.04$ | $\mathbf{93.02} \pm 3.61$ | $\underline{91.80} \pm 2.84$ |
| 10 | $68.60 \pm 3.72$ | $70.60 \pm 4.54$ | $\underline{75.11} \pm 2.00$ | $\mathbf{76.80} \pm 1.38$ |
| 11 | $91.20 \pm 2.48$ | $90.20 \pm 1.47$ | $\underline{92.96} \pm 1.15$ | $\mathbf{94.80} \pm 2.57$ |
| Summary | $62.71 \pm 2.95$ | $63.90 \pm 1.95$ | $68.12 \pm 1.91$ | $73.47 \pm 1.82$ |

Table 4: Optimal hyperparameter settings.

| Hyperparameter | BCI | SSVEP | ERN |
|----------------|-----|-------|-----|
| Batch Size | 32 | 32 | 32 |
| Learning Rate | $1e^{-3}$ | $1e^{-3}$ | $1e^{-4}$ |
| Weight Decay | $1e^{-2}$ | $1e^{-2}$ | $1e^{-3}$ |
| Windows | 5 | 1 | 4 |
| LoRA Learning Rate | $1e^{-2}$ | $1e^{-5}$ | $1e^{-1}$ |

