# OpenReview forum: "LAtte: Hyperbolic Lorentz Attention for Cross-Subject EEG Classification"
_ICLR.cc/2026/Conference — ICLR 2026 Conference Withdrawn Submission_

### Official Review · Reviewer_kDw8 · 2025-10-30

**Soundness:** 2
**Presentation:** 2
**Contribution:** 2
**Rating:** 4
**Confidence:** 4

**Summary:**

This paper proposes a fully hyperbolic neural network for cross-subject EEG classification, combining Lorentz attention, a hyperbolic InceptionTime encoder, and LoRA-based subject embedding. However, the claimed hierarchical nature of EEG data lacks empirical justification, the evaluation protocol is under-specified and recent baselines are missing.

**Strengths:**

Novel use of fully hyperbolic architecture tailored for EEG.

**Weaknesses:**

1. Unsubstantiated Hierarchy Claim

The paper makes a strong claim that EEG data has a hierarchical structure inherently suited for hyperbolic space. However, no empirical evidence is provided to support this assertion. References to the spatial arrangement of EEG sensors are insufficient, as sensor layout does not imply a latent hierarchy in the underlying data manifold.

2. Missing Comparisons with Recent Foundation Models

This work lacks comparison with recent high-impact foundation models such as BIOT (NeurIPS' 2023) [1], LaBraM (ICLR' 2024) [2], and CBraMod (ICLR' 2025) [3]. These models are widely cited, have open-source code and model weights provided, and achieve state-of-the-art results on cross-subject EEG decoding tasks. In contrast, the comparison baselines in this paper are largely outdated, originating from work published 3–5 years ago.

3. Unclear Evaluation Protocol and Potential Fairness Concerns

The evaluation protocol is under-specified. It remains unclear whether the authors used leave-one-subject-out (LOSO), random subject splits, or session-based splits. The fine-tuning strategy also lacks important details, for instance, the proportion of each subject's data used for fine-tuning versus testing.
Although the authors state on Page 8 (lines 416–419) that they follow the protocol of Pan et al. (2022), with session-wise splits for SSVEP and ERN, and instance-wise splits for MI, these strategies are not equivalent to true cross-subject evaluation.
Moreover, Section 4.6 describes a "subject-conditional fine-tuning" approach, but fails to specify how much subject data is used for adaptation, raising reproducibility concerns.

Importantly, CBraMod [3] uses the same BCI-IV-2a (MI) dataset and reports stronger results (see Table 23) using a clearly defined LOSO protocol. Without ensuring consistent evaluation settings, performance comparisons may be unfair or misleading.


Refs:

[1] Yang, Chaoqi, M. Westover, and Jimeng Sun. "Biot: Biosignal transformer for cross-data learning in the wild." Advances in Neural Information Processing Systems 36 (2023): 78240-78260.

[2] Jiang, Weibang, Liming Zhao, and Bao-liang Lu. "Large Brain Model for Learning Generic Representations with Tremendous EEG Data in BCI." The Twelfth International Conference on Learning Representations.

[3 Wang, Jiquan, et al. "CBraMod: A Criss-Cross Brain Foundation Model for EEG Decoding." The Thirteenth International Conference on Learning Representations.

**Questions:**

See comments above.

---

### Official Review · Reviewer_afFU · 2025-11-01

**Soundness:** 2
**Presentation:** 1
**Contribution:** 2
**Rating:** 2
**Confidence:** 4

**Summary:**

This paper introduces LAtte, a novel framework for cross-subject EEG classification that addresses the challenges of low signal-to-noise ratio (SNR) and high inter-subject variability inherent in EEG data. The core of LAtte is a fully hyperbolic model that integrates a Lorentz Attention Module with an InceptionTime-based encoder. Unlike much prior work focusing on single-subject performance, LAtte is explicitly designed for cross-subject generalization.

**Strengths:**

1.	The LoRA-based subject adapters offer a lightweight way to encode subject identity and separate subject-specific noise distributions while still training a single joint model. This addresses a frequent bottleneck in real BCI workflows (per-subject retraining).
2.	Explicitly moving beyond single-subject (SS) training to focus on the more clinically relevant subject-conditional (SC) setting is a strong point
3.	LAtteJoint reports state-of-the-art results in the SC setting across all three datasets (MI, SSVEP, ERN).

**Weaknesses:**

1.	The discussion of related work reads more like a list of prior studies rather than a coherent synthesis. The relationships among different works are unclear, and the authors fail to explicitly connect the cited literature to the motivation or design choices of their own model. This section requires substantial restructuring to establish a logical narrative.
2.	The innovation of this paper is quite limited. Its components (such as the Lorentz attention mechanism and hyperbolic functional connectivity) largely involve applying known hyperbolic building blocks to EEG data. No novel approaches are presented in either pretraining or fine-tuning strategies.
3.	Cross-subject generalization to unseen subjects is asserted but not isolated. The SC setup uses subject metadata during training; it’s not fully clear whether leave-one-subject-out (train on S-1, test on held-out subject with only ID available) is performed and reported distinctly from joint-training with all subjects.
4.	While 10 seeds are used and per-subject SSVEP tables are provided in the appendix, the main tables lack statistical significance testing across methods.
5.	The paper mixes (a) reconstruction and (b) cut-and-fill as pretraining tasks and later uses a cut-only variant as augmentation; the exact schedule is under-specified.
6.	The experimental comparison omits several strong baselines such as EEG Conformer and ATCNet. Compared to the current true state-of-the-art, LAtte's performance is about 10% lower, which substantially weakens the claimed contribution and significance of the work.

**Questions:**

Please refer to the Weaknesses section for detailed questions and suggestions to the authors.

---

### Official Review · Reviewer_c2j3 · 2025-11-01

**Soundness:** 2
**Presentation:** 3
**Contribution:** 3
**Rating:** 2
**Confidence:** 4

**Summary:**

LAtte presents the first fully hyperbolic EEG classification architecture operating entirely in Lorentz space, extending hyperbolic modeling beyond the decoder to the complete pipeline. The work addresses genuine challenges in cross-subject EEG classification through principled geometric design combined with LoRA adapters. Both reviews recognize the novelty and potential impact of this contribution. However, significant concerns exist around experimental validation completeness (missing foundation model baselines, LOSO evaluation, noise robustness experiments), statistical rigor (no significance tests), and presentation clarity. The synthesis recommends Borderline Accept with substantial revisions required to strengthen empirical validation and comparative positioning relative to recent EEG foundation models.

**Strengths:**

1. **Novel Architectural Contribution:** Both reviews recognize LAtte as the first fully hyperbolic pipeline for EEG classification, representing a meaningful step beyond prior partially hyperbolic approaches.
2. **Cross-Subject Focus Addresses Important Problem:** The cross-subject generalization motivation is genuine and well-articulated.
3. **Comprehensive Ablation Studies:** The ablation study systematically validates component contributions.
4. **Presentation Has Accessibility Strengths:** Geometric concepts are introduced accessibly.

**Weaknesses:**

1. **Missing Foundation Model Baselines:** Recent foundation-style EEG models (CebraMod, LaBraM) demonstrate strong cross-subject transfer and state-of-the-art performance Without these comparisons, cannot determine if hyperbolic design advantages outweigh data-driven massive pretraining *Impact:* Undermines competitive positioning claims and makes it unclear whether architectural innovation provides value beyond scale
2. **LOSO Validation Missing:** Standard protocol for validating true cross-subject generalization to completely unseen subjects Current session-wise/instance-wise splits include data from all subjects during training *Impact:* Cross-subject generalization claims are not supported by strongest evaluation protocol
3. **Statistical Significance Tests Absent:** No significance tests to validate whether improvements are statistically meaningful Critical gap for publication standards *Impact:* Cannot determine if performance gains are reliable or due to chance
4. **Noise Robustness Not Empirically Evaluated:** Low SNR is central motivation for using hyperbolic geometry No experiments directly test robustness (noise injection, artifact-heavy recordings, low-quality datasets) *Impact:* Core motivation not validated experimentally
5. **Computational Cost Not Analyzed:** No runtime, memory, or FLOPs comparison provided Efficiency claims regarding JL projection not supported with quantitative evidence *Impact:* Practical applicability unclear, efficiency contribution overstated

**Questions:**

1. **Add Foundation Model Baselines:** Compare against CebraMod (Wang et al., 2024) and LaBraM (Jiang et al., 2024) Use same datasets and evaluation protocols for fair comparison
2. **Conduct LOSO Evaluation:** Report leave-one-subject-out cross-validation results Demonstrate true generalization to completely unseen subjects
3. **Add Statistical Significance Tests:** Conduct paired t-tests or Wilcoxon signed-rank tests across folds/subjects Report p-values and effect sizes for claimed improvements
4. **Empirically Validate Noise Robustness:** Design noise injection experiments at varying SNR levels Test on artifact-heavy recordings or low-quality datasets

**Details Of Ethics Concerns:**

None.

---

### Official Review · Reviewer_SuoM · 2025-11-08

**Soundness:** 2
**Presentation:** 2
**Contribution:** 2
**Rating:** 4
**Confidence:** 3

**Summary:**

This paper introduces LAtte, a framework for cross-subject EEG classification that integrates a Lorentz Attention Module with an InceptionTime-based encoder, operating in hyperbolic space. The approach addresses key challenges in EEG data, such as low signal-to-noise ratio and inter-subject variability, by pretraining to learn shared baseline signals across subjects and using low-rank adapters (LoRA) to capture subject-specific differences.

**Strengths:**

The hyperbolic design extending prior work by incorporating hyperbolic operations in both encoder and decoder. It leverages hyperbolic geometry to better model hierarchical structures in EEG data.

LAtte's use of LoRA for subject embeddings allows joint training across subjects while maintaining adaptability.

The self-supervised pretraining tasks effectively handle noisy EEG data, and the Johnson-Lindenstrauss-inspired projection adds efficient regularization. Ablation studies demonstrate the effectiveness of each component

The authors made code available to review, and also commits to releasing code, pretrained checkpoints, which contributes to reproducibility.

**Weaknesses:**

The motivation for hyperbolic space is sound, but analysis of why it suits EEG hierarchies is needed, which is currently missing.

The paper lacks discussion on training/inference time, parameter counts, or scalability compared to baselines. Hyperbolic operations can be computationally intensive, so benchmarching could be useful.

For sections such as the Lorentz operations and attention mechanism, additional intuitive explanations on motivations behind the design and related equations would be beneficial.

No statistical significance tests on improvements, high per-subject variance isn't well mitigated following the experiment result.

**Questions:**

Could we visualized resulted hyperbolic embeddings from the experiment?

---

### Note · Authors · 2025-11-27

**Comment:**

We thank the reviewers for their feedback and will integrate the suggested changes for a resubmission.

**Withdrawal Confirmation:**

I have read and agree with the venue's withdrawal policy on behalf of myself and my co-authors.